# Discrepancy Study of the Chemical Constituents of Panax Ginseng from Different Growth Environments with UPLC-MS-Based Metabolomics Strategy

**DOI:** 10.3390/molecules28072928

**Published:** 2023-03-24

**Authors:** Yizheng Sun, Xiaoyan Liu, Xiaojie Fu, Wei Xu, Qingmei Guo, Youbo Zhang

**Affiliations:** 1School of Pharmacy, Shandong University of Traditional Chinese Medicine, Jinan 250355, China; 2State Key Laboratory of Natural and Biomimetic Drugs, School of Pharmaceutical Sciences, Peking University, Beijing 100191, China; 3Department of Natural Product Chemistry, Key Lab of Chemical Biology of Ministry of Education, School of Pharmaceutical Sciences, Shandong University, Jinan 250012, China

**Keywords:** *P. ginseng*, ginsenosides, growth environment, metabolomics, multivariate statistical analysis, UPLC-MS/MS

## Abstract

Panax ginseng (*P. ginseng*), the dried root and rhizome of *P. ginseng* C. A. Meyer, is widely used in many fields as dietary supplements and medicine. To characterize the chemical constituents in *P. ginseng* cultivated in different growth environments, a UPLC-TOF-MS method was established for qualitative analysis. Four hundred and eight ginsenosides, including 81 new compounds, were characterized in *P. ginseng* from different regions. Among the detected compounds, 361 ginsenosides were recognized in *P. ginseng* cultivated in the region of Monsoon Climate of Medium Latitudes, possessing the largest amount of ginsenosides in all samples. Furthermore, 41 ginsenosides in 12 batches of *P. ginsengs* were quantified with a UPLC-MRM-MS method, and *P. ginsengs* from different regions were distinguished via chemometric analysis. This study showed that the different environments have a greater influence on *P. ginseng*, which laid a foundation for further quality control of the herb.

## 1. Introduction

Ginseng, mainly cultured in Northeast Asia, usually refers to the dried root and rhizome of *P. ginseng* C. A. Meyer (*P. ginseng*) [1], and it has been regarded as one of the greatest elixirs in China for centuries. Due to its efficacy and lower adverse effects, *P. ginseng* is popular in many countries around the world. As a commonly used functional food, dietary supplement and drug in clinical worldwide [2,3,4,5], ginseng has a variety of beneficial effects, such as anti-aging, anti-cancer, anti-diabetes, immunomodulation, neuromodulation [6,7,8,9,10], antiatherosclerosis [11,12], neuroprotection [13,14,15,16], anti-inflammation [17], anti-melanin [18], and treatment of chronic obstructive pulmonary disease [19] and organ fibrosis [20,21]. Thus, it is important to define the kinds and contents of active ingredients in *P. ginseng*. Due to the chemical diversity in Chinese herbal medicine, accurate analysis of these metabolites is still a great challenge. Benefiting from advanced liquid chromatography (LC) coupled with mass spectrometry (MS) technology, a novel method was used to characterize and quantitatively analyze ginsenosides in *P. ginseng* from different growth environments.

Wild and cultivated *P. ginseng* traditionally grow in a limited area of monsoon climate of medium latitudes (127° E–128° E, 41° N–42° N). Nowadays, the cultivation areas of *P. ginseng* are widespread and have expanded to other areas, such as the subtropical monsoon climate area. In recent years, *P. ginseng* has been widely invested because of the continuous expansion of ginseng cultivation areas and the increasing status of ginseng in medicinal herbs. However, not all regions can provide a high-quality environment for the growth of ginseng; the expansion of cultivation regions without choice is obviously unscientific. Different latitudes and longitudes receive different light, moisture, and temperature, which may result in the diversity of metabolites in *P. ginseng* grown in different regions, especially in areas with large climate and terrain differences.

Ginsenosides are the representative compounds of *P. ginseng*. The Chinese Pharmacopoeia uses ginsenosides as indicators to evaluate the quality of *P. ginseng*, which suggests the importance of ginsenosides in *P. ginseng* quality control. Currently, more than 200 ginsenosides have been reported, which were divided into four categories: protopanaxadiol (PPD), protopanaxatriol (PPT), oleanolic acid (OA), and pseudo (PG) types [22,23]. However, the market today is popular with *P. ginseng* from many sources, and their prices are very different. This may be attributed to the ginsenosides in *P. ginseng* [24]. Therefore, it is of great significance and urgency to investigate the chemical constituents of *P. ginseng* and their correlation with the quality of the drug in different regions. A precise targeted and untargeted research on ginsenosides was operated in this study.

Benefiting from modern technology, high-resolution mass spectrometry, such as Quadrupole Time of Flight (Q-TOF), has been maturely applied to the non-targeted study of metabolites without reference [25,26,27,28,29]. In addition, liquid chromatography-electrospray ionization triple-quadrupole tandem mass spectrometry (LC-ESI-QQQ-MS/MS) coupled with multiple reaction monitoring (MRM) is also a mature technique for marker-targeted quantitative analysis [30,31]. However, there are few overall studies on targeting and non-targeting of metabolites at the same time, resulting in the inability to fully reveal the characteristics of a particular medicinal material. Hence, a rapid and sensitive UPLC-TOF-MS/MS combined UFLC-ESI-MS/MS method for the targeting, and non-targeting analysis of ginsenosides was established to explore the discrepancy of the metabolites of *P. ginseng* from different latitudes and longitudes [24].

Consequently, the aim of this study was to combine non-targeted and targeted methods for metabolite characterization and quantitatively analyze *P. ginseng* from different latitudes and longitudes. For non-targeted analysis, the aglycons, glycosyl or acyl groups of these ginsenosides were identified by the retention time, precursor ion, and fragment ions of each compound. Then, the deduced molecular formula was searched in the SCI-Finder database. Totally 408 ginsenosides were identified from different *P. ginseng* samples and 81 potential new ginsenosides were identified. For targeted analysis, 41 ginsenoside markers in four groups of *P. ginsengs* were quantified by a rapid and accurate UPLC-MS/MS method. Furthermore, multivariate statistical methods were used for quantitative analysis, aiming to distinguish ginseng from different geographical locations by ginsenosides. The results showed that the Latitudes and longitudes have an important effect on the chemical constituents’ composition and content of *P. ginseng*.

## 2. Results and Discussion

### 2.1. Analysis of Untargeted Potential Metabolites

A total of 408 ginsenosides, including 81 new potential compounds, were detected in the four groups of *P. ginseng*. MS/MS data of the new and known ginsenosides are separately shown in Appendix A [32,33,34,35,36,37,38,39,40,41,42,43,44,45,46]. As shown in Appendix A, up to 16 aglycones derived from protopanaxadiol (PPD), protopanaxatriol (PPT), oleanolic acid (OA), and pseudo-ginsenoside (PG), five glycosyl groups and 17 acyl groups were identified in the 408 ginsenosides. After careful comparison, we found that the references of DEDT (48) and DDT (59) could only be detected in the positive ion mode, while the detection of other ginsenosides was set in the negative mode due to better peak shapes and sensitivity. A total of 153 PPD-type ginsenosides, 210 PPT-type ginsenosides, 9 PPD or PPT-type ginsenosides, 13 OA-type ginsenosides, and 29 PG-type ginsenosides were detected and identified from all samples cultivated in the four areas. The PPD-type and PPT-type ginsenosides account for the largest proportion of ginsenosides in our research, similar to relevant studies on *P. ginseng* [22]. The 81 new ginsenosides were recognized and divided into four types: 41 PPT-type ginsenosides, 29 PPD-type ginsenosides, 4 PG-type ginsenosides, 1 OA-type ginsenosides, and six ginsenosides that could not be accurately determined as PPD or PPT-types. The structural classification of the 81 new ginsenosides is shown in Figure 1.

### 2.2. The Fragmentation Regularity of Untargeted Potential Metabolites

As shown in Appendix A. PPD1-A is the aglycone of protopanaxadiol ginsenosides, while PPD1-B is the isomer of PPD1-A with C_20_–_22_ dehydration and C_25_ hydroxylation that shares the same ion with PPD1-A at *m*/*z* (459). The remaining PPD types are the compounds with side chain changes of PPD. PPD2-A and PPD2-B are C_20_–C_22_-dehydrated compounds of PPD1-A and share the same ion at *m*/*z* 441. The difference between the two types of ginsenosides is the olefinic bond position at the side chain. PPD3, with fragment ion at *m*/*z* 487, is the C_20_–C_22_-dehydrated-, C_23_-methoxylated-, and C_24_–C_25_-epoxidated ginsenoside of PPD1-A. PPD4 is the C25-hydroxylated PPD1-A, showing its fragment ion at *m*/*z* 477. PPD5 is the C_24_–C_25_-hydrogenated PPD1-A, and its characteristic fragment ion is at *m*/*z* 461. PPT1 is the aglycone of protopanaxatriol ginsenosides, and the remaining PPT types are compounds with side chain changes. PPT2-B (*m*/*z* 457) and PPD2-A are isomers and share the same fragment ion at *m*/*z* 475. The two types of ginsenosides are C_20_–C_22_-dehydrated ginsenosides of PPT1 with different positions of the olefinic bond at the side chain. PPT3-type (*m*/*z* 473) ginsenosides are the C_20_–C_22_-dehydrated and C_24_–hydroxylated compounds of PPT1, with a switched olefinic bond from C_24_–C_25_ to C_25_–C_26_. PPT4 (*m*/*z* 493) is the C_25_-hydroxylated ginsenosides of PPT1. PPT5-A and PPT5-B are isomers at *m*/*z* 477; their difference was the position of a hydroxyl group (C_25_-OH of PPT5-A and C_20_-OH of PPT5-B). These two ginsenosides types are C_24_–C_25_-dehydrated compounds of PPT1. The ions at *m*/*z* 337, 279 are the evidence of aglycone OA (*m*/*z* 455), while the ions at *m*/*z* 391, 257 are the evidence of aglycone PG (*m*/*z* 491). The types and linkage of glycogens and acyls are identified by neutral fragments. Finally, the molecular formulae of all ginsenosides are obtained by precise analysis of prerequisite ions and MS^2^ fragments derived from specific prerequisite ions and retention times.

As examples of compound identification, the secondary mass spectra of C33 and C42 and the fragmentation process are shown in Figure 2. The molecular formula of C33 was confirmed as C_55_H_92_O_19_ with [M-H]^−^ at 1044.6233. The mass spectrum confirmed that the aglycone of C33 (t_R_ = 8.48 min) was PPD1-A which showed characteristic ions at *m*/*z* 459, 375. From the MS^2^ spectrum, the difference between *m*/*z* 1043.5402 [M-H]^−^, 945.5465 [M-H-hexanoyl]^−^, 783.4928 [M-H-hexanoyl-glc]^−^, 621.4458 [M-H-hexanoyl-2glc]^−^, and 459.3973 [M-H-hexanoyl-3glc]^−^ indicated the successive loss of a hexanoyl and three glucosyl groups. Thus, this compound was named as hexanoyl-Rd (Figure 2A). The [M+FA-H]^−^ and [M-H]^−^ of C42 were at *m*/*z* 833.4575 and 787.5343, respectively. In the MS^2^ spectrum of C42, the difference between 787.5343 [M-H]^−^, 637.4324 [M-H-decadienoyl]^−^, and 475.3793 [M-H-decadienoyl-glc]^−^ indicated the presence of a decadienoyl and a glucosyl groups. The characteristic ions at *m*/*z* 475 and 391 in the MS^2^ spectrum confirmed the aglycones of C42 (t_R_ = 10.56 min) were PPT1-type. It was concluded that the molecular formula of C42 was C_46_H_76_O_10_ at 788.5438. Thus, this compound was named decadienoyl-Rh1 (Figure 2B).

### 2.3. Chemometrics Analysis for 41 Targeted Compounds

Based on the untargeted analysis, we performed an accurate quantitative analysis of 41 marker standards for the same *P. ginseng* samples. The same mobile phase gradient as in the untargeted study was used for UPLC-MS/MS analysis. The contents of 41 marker standards in 4 *P. ginseng* batches are shown in Appendix A. The content histogram of each analyte is shown in Appendix A, and the MRM mass spectrograms of each group of samples are shown in Appendix A. The results showed that PG-F_11_ (6) and PG-RT_5_ (7) could not be detected in all batches. Analytes 3, 4, 5, 8, 17, 21, 20 and 22 were higher than other analytes and were the main ginsenosides in *P. ginseng*. The other analytes could be detected and quantitated from all *P. ginseng* batches, but their content of them was much lower. Additionally, the proportion of all ginsenosides was similar in B1 and B2. The content of G-Re (4) is the highest in B1 and B2. For B1, G-Rg_1_, G-Re, G-Ro, and G-Rb_3_ are the four main ginsenosides, while G-Re, G-Ro, CS-Iva, and G-Rb_3_ are the four main ginsenosides in B2. The figure for B3 showed that the content of G-Ra_1_ is the highest, G-Rg_1_, G-Re, G-Ro, and G-Ra_1_ are the four most abundant ginsenosides. The content of different ginsenosides in B4 is much lower than in the other three groups, and even the main ginsenosides in B1, B2 and B3 also showed no obvious advantage in B4. Interestingly, PG-F_11_ could not be detected, which is consistent with the ginsenoside composition of Asian ginseng [47]. The quantitative analysis results showed that the content of the PPT ginsenosides type was the highest, and the PPD ginsenoside type was the second highest, which were the same as the result of untargeted potential metabolites analysis.

### 2.4. Discussion

As shown in Figure 3A, an OPLS-DA analysis was processed to analyze the differences between *P. ginseng* from different regions. The results showed that the parallelism of three parallel samples in each group was reliable. In addition, there were significant differences among the four groups, which indicated that the growing environment and region influenced the chemical composition of *P. ginseng*.

Figure 3B showed a heat map based on semi-quantitative results (shown in Appendix A) of the 408 ginsenosides. The cluster analysis in the heat map gathered B2 and B4 into a group and B1 and B3 into another group. Interestingly, compared with B3 and B4, B1 and B2 are closer to the ocean and have the same climatic conditions, terrain environment, longitude and latitude. However, the statistical analysis grouped B1 and B3 together, indicating that the environment is not the only criterion for *P. ginseng*. Both B1 and B3 grow at higher elevations, while B2 and B4 grow in areas with relatively gentle terrain. Thus, the topography may be an important environmental factor in the grouping, affecting the type and content of metabolites in *P. ginseng*. It is preliminarily speculated that the influence of the growth environment on the quantity and type of ginsenosides is greater than that of the *P. ginseng* growth region. The Venn diagram on all 408 ginsenosides is shown in Figure 3C. The number of saponins of *P. ginseng* from different regions is in the order of B1 > B3 > B2 > B4. The number of ginsenosides in B1 and B3 is higher than that of B2 and B4, respectively. The climatic conditions affected by different latitudes and longitudes are important factors affecting the metabolites in ginseng. B1 and B2 have higher latitudes and temperate monsoon climates. B3 and B4 have lower latitudes and are subtropical monsoon climates. Firstly, in terms of climatic characteristics. Due to the similar climate conditions of humid, the temperature is an important difference between the two. The temperature in the high latitude is lower than it is in the low latitudes, and *P. ginseng* prefers to grow in a cooler environment. Secondly, in terms of topography, there is more shade caused by the intricate shrubs in mountainous areas, and the blocking out of sunlight is beneficial to *P. ginseng* growth. The climate condition in mountainous areas is generally humid and has plenty of oxygen, which is more suitable for *P. ginseng* growth. Thirdly, for geographical location, the closer distance to the ocean ensures a wetter environment, which provides a more suitable environment for *P. ginseng* to grow than inland. 

According to Figure 3C, 257 ginsenosides were detected in all 4 *P. ginseng* groups. Additionally, there were unique ginsenosides in each group. Three unique ginsenosides for B4, nine unique ginsenosides for B3, seven unique ginsenosides for B1, and eight unique ginsenosides for B2, which showed the diversity and specificity of the chemical components in *P. ginseng* from different regions. The base peak chromatogram profiles and distribution of various types of ginsenosides in different samples are shown in Figure 4. The specific components in each group of the samples indicated the growth environment and growth area would affect the chemical composition of *P. ginseng*. 

Previous researchers have found that the types of ginsenosides in *P. ginseng* from different growth environments are obviously different, which also supports the results of this study [48,49]. The unique components in the four *P. ginseng* groups can be used as indicators to identify *P. ginseng* from different growth areas. The base peak chromatogram profiles and distribution of various types of ginsenosides in different samples are shown in Figure 4. The specific components in each group of the samples indicated the growth environment and growth region have effects on the chemical composition of *P. ginseng*. In addition, the composition of various ginsenoside types in each group is similar, of which the proportion of PPT-type ginsenosides was the largest and the amount of PPD-type ginsenosides was slightly less than PPT-type, and the content of PPD-and PPT-type in the tested samples was much higher than other ginsenoside types (Figure 3D). This is consistent with the characteristic of *P. ginseng* in the ratio of the chemical composition of PPT and PPD ginsenoside types [22]. The amounts of OA and PG were similar, but the quantity of PG was higher than that of OA. Of note, PG-F_11_ is a unique compound that only exists in American ginseng [47]. One of its isomers was detected at t_R_ = 5.01 min in our *P. ginseng* samples, which has similar fragmental ions with PG-F_11_.

The PCA and OPLS-DA analysis was established to perform the similarities and differences analysis in 12 sources of *P. ginseng* from various regions. As shown in Figure 5A,B, 12 sources of *P. ginseng* samples were divided into four groups (B1, B2, B3, and B4); there is no clear demarcation between B1 and B2 in supervised and unsupervised cluster analysis. In addition, there are clear boundaries between them and the other groups (B3 and B4). 

As shown in Figure 5C, the heat map showed that the contents of different ginsenosides in B1 and B2 were similar, whereas the chemical composition of each group was different. The total content of the 41 analytes and the proportion of various ginsenoside types in B1, B2, B3, and B4 were calculated and analyzed. The results showed that the content of total ginsenosides in B1, B2, and B3 was much higher than that in B4 (Figure 5D). Since the chemical composition of B1 and B2 were closely related, we compared the proportion of different ginsenoside types of them. In Figure 5D, the proportion of PPT- and PPD-type of B1 were significantly higher than those of B2. However, the proportion of OA-type ginsenosides in B1 was lower than it was in B2. It is reasonable that the quality of *P. ginseng* from B1 is better than *P. ginseng* from B2, which is due to the advantage of higher terrain with less sunlight and sufficient water. Additionally, the PPD-type and OA-type in B3 were lower than those of B1, and all ginsenoside types in B4 were lower than those in other groups. As for B3 and B4, the lower latitude is associated with higher temperatures, which is not suitable for *P. ginseng* growth. Meanwhile, the higher longitude for B4 means that the growing environment of *P. ginseng* is far away from water sources, and the plain area may provide more sunlight; these factors may be the fundamental reasons for the generally low ginsenosides content in B4. Based on these results, the ginsenosides content and composition of *P. ginseng* were determined by the combination of several environmental factors, including the latitude and longitude and climate conditions.

## 3. Experimental

### 3.1. Materials

As shown in Figure 6, 12 batches of *P. ginseng* samples were obtained from 4 different regions (B1–B4, detailed information is shown in Appendix A). All *P. ginseng* samples were harvested at five years. B1 (The hills of a plain region, Monsoon Climate of Medium Latitudes) and B2 (Plain, Monsoon Climate of Medium Latitudes) were collected from 41°41′ N–42°25′ N, 127°42′ E–128°16′ E. B3 was from 26°59′ N–27°32′ N, 103°09′ E–103°40′ E (Vysocina, Subtropical monsoon climate), and B4 was collected from 29°01′ N–33°06′ N, 108°21′ E–116°07′ E (Plain, Subtropical monsoon climate). All the voucher specimens, identified by Professor Xiuwei Yang of Peking University, were stored in the State Key Laboratory of Peking University.

### 3.2. Standard Samples, Chemicals, and Reagents

As shown in Appendix A, 63 ginsenosides, with purities above 98%, were used as the standards for non-targeted analysis. Acetonitrile, methanol, and ammonium formate suitable for LC-MS analysis were purchased from Thermo Fisher Scientific (Fair Lawn, NJ, USA). Water was produced by Millipore Alpha-Q Water Purification System (Bedford, MA, USA).

### 3.3. Preparation of Standard Solution and Samples

Each ginsenoside reference was weighed accurately and dissolved in methanol. A mixed standard reference solution was prepared for LC-MS analysis. 0.5 g *P. ginseng* powder (60 mesh size) of each sample was extracted by 20 mL 70% MeOH for 30 min in ultrasonic apparatus (250 W, 40 kHz). The standard and extracted solution were stored at −20 °C and filtered through a 0.22 μm filter membrane before analysis.

### 3.4. Instruments, Conditions, and Parameters of Analysis

The analysis was performed on a SCIEX triple TOF 6600+ mass spectrometer system equipped with the SCIEX Exion LC AD system. An ACQUITY UPLC ^®^ CSH (C18 1.7 μm, 2.1 ×100 mm, Waters) with an AC quality UPLC ^®^ BEH Shield RP-18 1.7 μm pre-column was used for separation. The mobile phase was composed of 0.5 mM ammonium formate in water (A) and acetonitrile (B). The gradient scale was as follows: 0–3 min 22% B; 3–5 min 22–30% B; 5–9 min 30–35% B; 9–11 min 35–40% B; 11–13 min 40–48% B; 13–18 min 48–55% B; 18–22 min 55–70% B; 22–25 min 70–90% B; 25–26 min 90% B. The injection volume was 2 μL. The column oven was set at 40 °C. The automatic sampler was set at 25 °C, and the flow rate was 0.35 mL min^−1^. The MS/MS data detection was in both positive (ion spray voltage: 5500 V) and negative (ion spray voltage: −4500 V) ion acquisition modes. The parameters of the mass spectrometer were set as follows: the scanning range *m*/*z* 100–1500, the temperature of the ion source was set at 600 °C, the collision energy was 40 ± 20 V, and the declustering potential was −80/80 V.

### 3.5. Method of Non-Targeted Analysis

The mass data were processed by Peakview 1.2 software. Based on the MS and MS/MS fragmental data of the 63 references, the molecular formula and linkage of aglycones and glycosyls of the ginsenosides were, in turn, deduced according to their retention time, precursor ions and MS/MS fragments. The errors within 10 ppm between the measured and theoretical molecular weight of all ginsenosides were considered reasonable. The chemical structures of the identified compounds were searched in the SciFinder database to determine whether they were new ginsenosides. Subsequently, a metabolomics strategy was used to compare the differences in the chemical composition of *P. ginseng* samples from different regions. After centroiding, deisotoping, filtering, peak recognition and integration, a multivariate data matrix including sample identity, ion identity (t_R_ and *m*/*z*), and ion abundance was submitted to XCMS software for cluster and Venn analysis. Next, the multivariate data matrix was further analyzed with SIMCA-P 13.0 software (Umetrics, Kinnelon, NJ, USA). An OPLS-DA method was applied to obtain loading scatter S-plots. The ions that were responsible for the distinguishment of the different groups were selected and determined with their MS/MS data and the authentic standards.

### 3.6. Quantitative Analysis of Marker References

Targeted analysis for the same ginsengs as the non-targeted analysis was performed on LC-MS. The LCMS-8050 Triple Quadrupole Liquid Chromatograph Mass Spectrometer system (Shimadzu Corp., Kyoto, Japan) with equipment of electrospray ionization source (ESI) was used for the analysis of 41 marker references. The Ultra-Fast liquid chromatography (UFLC) system consists of a binary pump (LC-30AD), a column oven (CTO-20AC) and a prominence autosampler (SIL-30AC). The analysis conditions, including mobile phase, injection volume, flow rate, column temperature, autosampler temperature, and gradient scale, were the same as non-targeted analysis. The conditions of ESI-MS are based on the previous research of our research group [50], which are as follows: interface voltage, 3 kV; detector voltage, 2.6 kV; interface temperature, 300 °C; DL temperature, 250 °C; heating gas flow, 10 L/min; the flow rate of drying gas and nebulizer gas was 10 L/min and 3 L/min, respectively. The targeting analysis was performed in multiple reaction monitoring (MRM) modes under negative ion mode.

## 4. Conclusions

A rapid and sensitive targeted and non-targeted method for qualitative and quantitative analysis of ginsenosides has been successfully applied in this study. A total of more than 400 compounds, including 81 new compounds, have been identified in 12 sources of *P. ginseng* from four different geographies. C33 (named hexanoyl-Rd) and C43 (named decadienoyl-Rh1) are demonstrated for the process of compound fragmentation, which enriches the chemical library of *P. ginseng* and describes a novel method for saponin identification in this study. The structures of potential compounds were accurately identified by UPLC-TOF-MS based on retention time, precursor ions, and fragmentation. The targeted analysis of 41 ginsenosides directly indicated the effect of the growth environment on ginsenosides. *P. ginseng* grown in different geography has different chemical diversity, and the difference in topography, climate, moisture and sunlight caused by different latitudes and longitudes plays a key role in the composition and content of ginsenosides. The content of G-Ra1 increased with climate warming, and the content of G-Rg1 was positively correlated with altitude. CS-IVA accumulated more in plain-grown ginseng. In vivo accumulation of G-Re and G-Ro does not appear to be closely related to geographical location. In addition, with the growing popularity of dietary supplements and functional foods, *P. ginseng* will take a bigger share of the market as an important raw material for functional food and cosmetics. This experiment analyzed the differences in ginseng medicinal materials under different growth environments from the perspective of chemical components, though the relationship between the differences in chemical components and drug efficacy needs to be verified in future pharmacological experiments. 

## Figures and Tables

**Figure 1 molecules-28-02928-f001:**
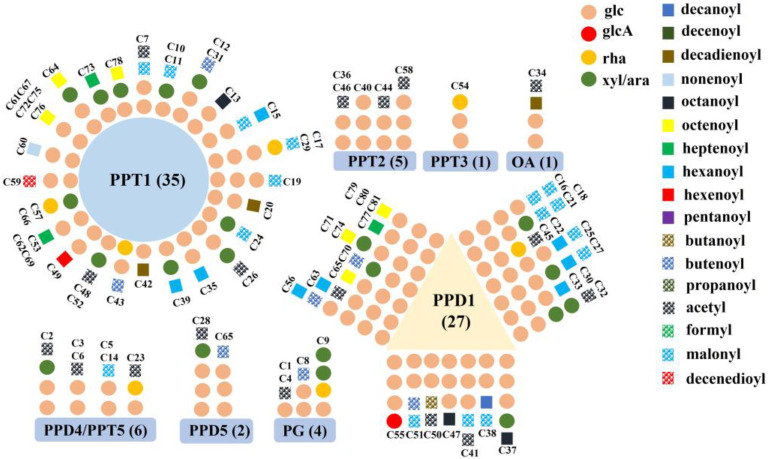
Schematic structures of the 81 new ginsenosides.

**Figure 2 molecules-28-02928-f002:**
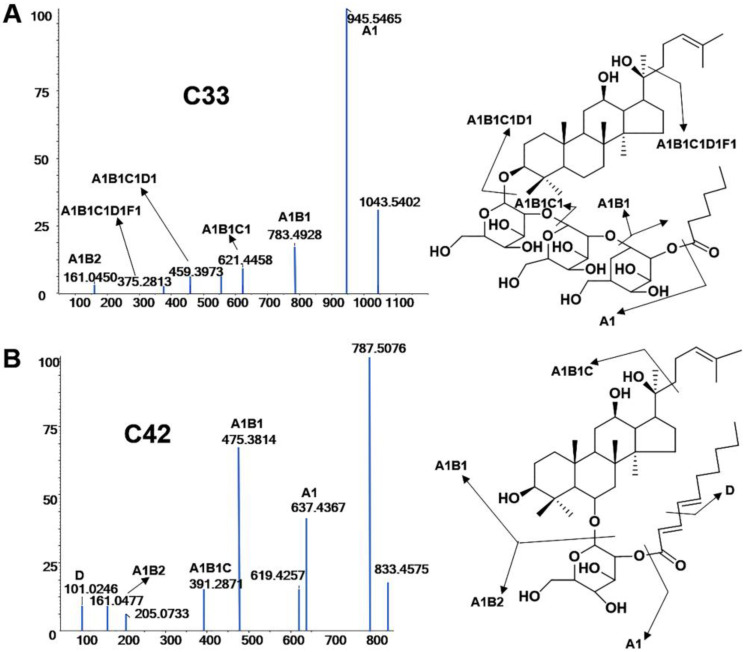
The secondary mass spectra and the fragmentation process: (**A**) C33, (**B**) C42.

**Figure 3 molecules-28-02928-f003:**
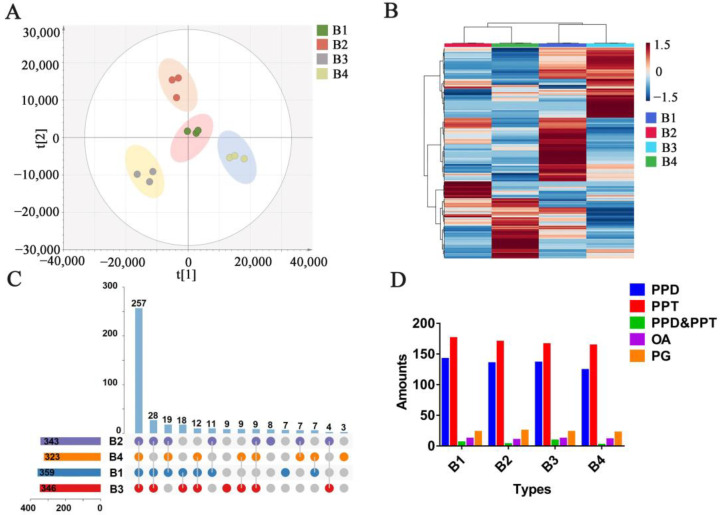
Multivariate statistical analysis of the *P. ginseng* samples. (**A**) OPLS-DA analysis of 4 groups of *P. ginseng*, (**B**) heat map analysis of *P. ginseng*, (**C**) The Venn diagram of 4 groups of *P. ginseng*, (**D**) Number of different types of ginsenosides in *P. ginseng* samples.

**Figure 4 molecules-28-02928-f004:**
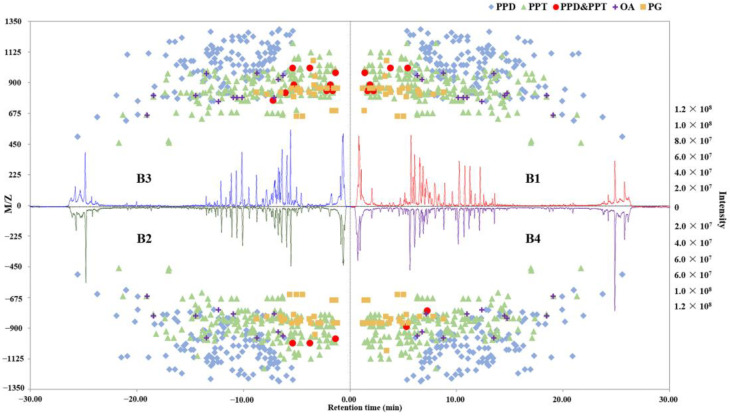
Base peak chromatogram profiles and distribution of various types of ginsenosides in *P. ginseng* samples.

**Figure 5 molecules-28-02928-f005:**
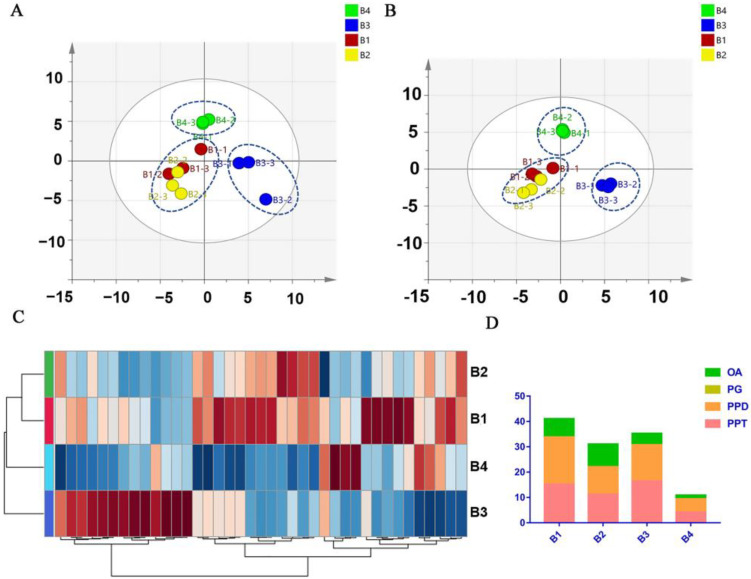
Multivariate statistical analysis of the *P. ginseng* samples. (**A**) PCA analysis. (**B**) OPLS-DA analysis. (**C**) heatmap. (**D**) The total content of each type of saponin.

**Figure 6 molecules-28-02928-f006:**
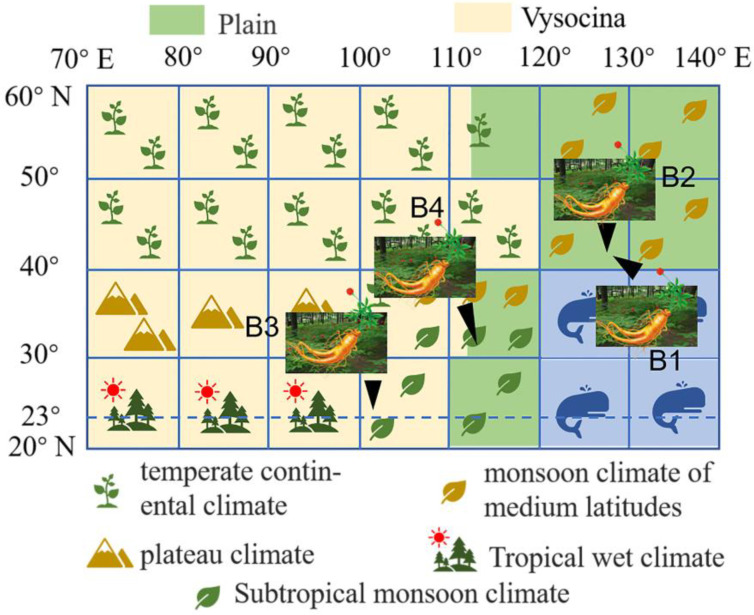
Geographic distribution of 4 groups of *P. ginseng* samples.

## Data Availability

Not applicable.

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
