# Peer review of "Discrepancy Study of the Chemical Constituents of Panax Ginseng from Different Growth Environments with UPLC-MS-Based Metabolomics Strategy"

_molecules, 2023, doi:10.3390/molecules28072928_

Round 1

Reviewer 1 Report

This manuscript aim to combine non−targeted and targeted methods for metabolites characterization and quantitatively analyze of P. ginseng from different latitudes and longitudes. For non−targeted analysis, the aglycons, glycosyl or acyl groups of these ginsenosides were identified by the retention time, precursor ion, and fragment ions of each compound. Then, the deduced molecular formula was searched in the SCI−Finder database. Totally 408 ginsenosides were identified from different P. ginseng samples, and 81 potential new ginsenosides were identified. For targeted analysis, 41 ginsenoside markers in four groups of P. ginsengs were quantified by a rapid and accurate UPLC−MS/MS method. Furthermore, multivariate statistical methods were used for quantitative analysis, aiming to distinguish ginseng from different geographical locations by ginsenosides. The results showed that the Latitudes and longitudes has an important effect on the chemical constituents’ composition and content of P. ginseng.

The work is meaningful with sufficient data and in line with readers' interests of Molecules. However, there are still some shortcomings that need to be further improved or explained.

-  I suggest separate results from Discussion, I mean if possible?.

- Line from 57- 72 need to add citations.

-   In introduction 12 references not enough.

-  There are a lot of taxonomic names either of the pathogens of marine sources that need to be in italics. Please edit this throughout.

- Conclusion very short for article contain this huge data.

-   References are much more important, the author should modify references and add the specific area references for example the author should cover 10-year work. Also 31 is not enough for research article, at increase to 50.

Author Response

Comment: I suggest separate results from Discussion, I mean if possible?

Response: We have checked the article and revised it according to your comments.

Comment: Line from 57- 72 need to add citations.

Response: We have added some references at this section.

Comment: In introduction 12 references not enough

Response: We have added some references in the Introduction section as your suggestion.

Comment: There are a lot of taxonomic names either of the pathogens of marine sources that need to be in italics. Please edit this throughout.

Response: We have checked the article thoroughly and revised them according to your comments.

Comment: Conclusion very short for article contain this huge data.

Response: Thank you for your comments. The conclusion has been enriched.

Comment: References are much more important; the author should modify references and add the specific area references for example the author should cover 10-year work. Also 31 is not enough for research article, at increase to 50.

Response: We have added some literatures that are closely related to this study.

Reviewer 2 Report

The work concerns the determination of ginsenosides in ginseng. The results add to our knowledge as the authors identified 81 new compounds. The authors showed very well that the variability of the environment affects the chemical composition of ginseng. The article will be interesting for readers. Prepared work is honest and transparent. However, there are linguistic errors. The reviewer submits his comments and questions below:

·        Line 31 replace „dietary supplement and drug” with dietary supplements and drugs

·        Line 32 replace „effects as anti−aging” with effects such as anti−aging

·        Line 35 replace „to defined” with to define

·        Line 62 replace „marker targeted” with marker-targeted

·        Line 69 replace „certain medicinal material” with particular medicinal material

·        Line 71 replace „to explored” with to explore

·        Lines 151-152 replace „differences of P. ginseng from different regions” with differences between P. ginseng from different regions

·        Line 175 replace „similarly” with similar

·        Line 179 replace „block out sunlight” with blocking out of sunligh

·        Line 189 replace „are showed” with are shown

·        Line 199 replace „each group are similar” with each group is similar

·        Lines 205-206 replace „This consistent” with This is consistent

·        Lines 208-209 replace „Of note, PG−F11 as a unique compound which only exists in American ginseng” with Of note, PG−F11 is a unique compound that only exists in American ginseng

·        Line 212 replace „accurate” with an accurate

·        Line 230 replace „were same” with were the same

·        Figure 5 should be placed below the text in which it is mentioned

·        Figure 6 should be placed below the text in which it is mentioned

·        Line 266 replace „information are shown” with information is shown

·        Line 305 replace „Chemical structures” with The chemical structures

·        Line 310 replace „ion abundance were submitted” with ion abundance was submitted

·        Line 320 replace „system consist of” with system consists of

·        Conclusions are written laconic. Please extend conclusions. Information on the names of selected novel compounds identified in ginseng may be added. Please also expand the information on the variable chemical composition of samples from different geographic locations.

·        Can differences between the chemical composition of individual samples allow us to conclude that some samples are better in terms of pro-health than others?

·        What are the further prospects of ginseng research?

·        Can the results obtained at work be used other than for product quality control? If so, please include this information in your conclusions

Author Response

Comment: Line 31 replace „dietary supplement and drug” with dietary supplements and drugs

Response: Thanks you for your question, the article has been revised accordingly.

Comment: Line 32 replace, effects as anti−aging” with effects such as anti−aging

Response: The sentence has been revised accordingly.

Comment: Line 35 replace „to defined” with to define

Response: We have revised the sentence as your suggestion.

Comment: Line 62 replace „marker targeted” with marker-targeted

Response: The words have been revised accordingly.

Comment: Line 69 replace „certain medicinal material” with particular medicinal material

Response: The sentence has been revised accordingly.

Comment: Line 71 replace „to explored” with to explore

Response: It has been revised accordingly.

Comment: Lines 151-152 replace „differences of P. ginseng from different regions” with differences between P. ginseng from different regions

Response: We have revised the sentence accordingly.

Comment: Line 175 replace „similarly” with similar

Response: The word has been revised as your suggestion.

Comment: Line 179 replace „block out sunlight” with blocking out of sunlight

Response: We have replaced the word “block” with “blocking”.

Comment: Line 189 replace „are showed” with are shown

Response: We have replaced the word “showed” with “shown”.

Comment: Line 199 replace „each group are similar” with each group is similar

Response: The word has been revised as your suggestion.

Comment: Lines 205-206 replace „This consistent” with This is consistent

Response: We have revised the sentence accordingly.

Comment: Lines 208-209 replace „Of note, PG−F11 as a unique compound which only exists in American ginseng” with Of note, PG−F11 is a unique compound that only exists in American ginseng

Response: We have revised the sentence as your suggestion.

Comment: Line 212 replace „accurate” with an accurate

Response: We have added the word “an” before “accurate”.

Comment: Line 230 replace „were same” with were the same

Response: We have revised the sentence as your suggestion.

Comment: Figure 5 should be placed below the text in which it is mentioned

Response: Figure 5 has been moved to the place below the text that describes multivariate statistical analysis of the P. ginseng samples.

Comment: Figure 6 should be placed below the text in which it is mentioned

Response: Figure 6 has been placed below the text that describes geographic distribution.

Comment: Line 266 replace „information are shown” with information is shown

Response: The word has been replaced.

Comment: Line 305 replace „Chemical structures” with The chemical structures

Response: We have added the word “The” before “chemical structures”

Comment: Line 310 replace „ion abundance were submitted” with ion abundance was submitted

Response: We have revised the word as your suggestion.

Comment: Line 320 replace „system consist of” with system consists of

Response: We have revised the mistake.

Comment: Conclusions are written laconic. Please extend conclusions. Information on the names of selected novel compounds identified in ginseng may be added. Please also expand the information on the variable chemical composition of samples from different geographic locations.

Response: Thank you for your suggestion. We have realized this question and enriched the content of conclusion section. Information of the selected novel compounds and the variable chemical composition of samples from different geographic locations were added and detailed described.

Comment: Can differences between the chemical composition of individual samples allow us to conclude that some samples are better in terms of pro-health than others?

Response: This study is chemical composition analysis of P. ginseng from different geographic locations. The chemical differences between the individual samples can help us to find P. ginseng with better quality. In terms of pro-health than others, better chemical composition may result to better pharmacological effect. While, we think the conclusion should be obtained when pharmacological research was also performed, and this is our further research in the next step.

Comment: What are the further prospects of ginseng research?

Response: P. ginseng has been used as one of the most distinguished elixir medicines for many centuries. The root of P. ginseng is considered as an important functional food and rich in ginsenosides. With the growing popularity of dietary supplements and functional foods, ginseng gradually became an important material in both the drug and food market. We believe that the results of this study will provide fundamental support for the quality control of ginseng. However, the effects of ginseng on human health, cosmetics industry, and dietary supplements still need to be further verified by experiments.

Comment: Can the results obtained at work be used other than for product quality control? If so, please include this information in your conclusions

Response: With the growing popularity of dietary supplements and functional foods, ginseng occupies the market as an important raw material for functional food and cosmetics. The results of this study provide accurate selection of medicinal origin for the application of ginseng in the market. However, pharmacological experiments are still needed to verify the effect of ginseng in clinical or functional food use.

Reviewer 3 Report

The manuscript "Characterization of ginsenosides from the root of Panax ginseng by integrating untargeted metabolites using UPLC−Triple TOF−MS" was focused on characterization of phytochemicals in Panax ginseng using untargeted and targeted metabolomic approaches. Through reviewing the manuscript, I have few minor points as follows:

- Could the authors add more information about the harvest year? Were these samples harvested in the same or different years?

- In Abstract the authors stated "408 ginsenosides including 81 new compounds were characterized in P. ginseng from different regions." while in Section 2.1, it mentioned 413 ginsenosides. Please explain.

Other than these, the results and discussion look fine.

Author Response

Comment: Could the authors add more information about the harvest year? Were these samples harvested in the same or different years?

Response: All the P. ginseng samples were harvested in the same year (5 years), and we have added the information in the text.

Comment: In Abstract the authors stated "408 ginsenosides including 81 new compounds were characterized in P. ginseng from different regions." while in Section 2.1, it mentioned 413 ginsenosides. Please explain.

Response: We apologize for the mistake. Total amount of the ginsenosides analyzed in the study is 408, and we have revised it in the manuscript.